# A generalized Knudsen theory for gas transport with specular and diffuse reflections

JianHao Qian ®[1], HengAn Wu ®[1,2] ✉ & FengChao Wang ®[1,2] ✉

Gas permeation through nanopores is a long-standing research interest because of its importance in fundamental science and many technologies. The free molecular flow is conventionally described by Knudsen theory, under the diffuse reflection assumption. Recent experiments reported ballistic molecular transport of gases, which urges for the development of theoretical tools to address the predominant specular reflections on atomically smooth surfaces. Here we develop a generalized Knudsen theory, which is applicable to various boundary conditions covering from the extreme specular reflection to the complete diffuse reflection. Our model overcomes the limitation of Smoluchowski model, which predicts the gas flow rate diverging to infinity for specular reflection. It emphasizes that the specular reflection can reduce the dissipation flow rate. Our model is validated using molecular dynamics simulations in various scenarios. The proposed model provides insights into the gas transport under confinement and extends Knudsen theory to free molecular flow with specular reflections.

Gas transport through nanochannels is a widespread phenomenon in nature and plays an essential role in various industrial fields, including membrane separation[1,2], gas extraction[3,4], nano catalysis[5] and vacuum technology[6,7]. In the regime of free molecular flow, the continuum theory breaks down and we have to rely on the kinetic theory of gases, including the Knudsen theory and the Boltzmann equation[8]. The Knudsen theory provides a concise approach to quantitatively describe the gas flow rate[9,10]. It was first purposed in 1909 by Martin Knudsen and its original formulation is only valid for a long channel with circular cross-section[11]. A year later, Smoluchowski derived an alternative corrected expression for long channels with arbitrary cross-section[12]. After that, this theory was further refined by Dushman[13], Clausing[14], etc., and has also been verified by extensive experiments[15–18]. The Knudsen theory gives the molar flow rate through a nanochannel connecting two gas reservoirs at pressure $P^{in}$ and $P^{out}$, in a form of

$$Q^K = \alpha_K \frac{\Delta PA}{\sqrt{2\pi RTM}},  \qquad (1)$$

in which $\Delta P = P^{in} - P^{out}$ is the pressure drop along the nanochannel with a cross-section area $A$, $R$ the gas constant, $T$ the temperature, and $M$ the molar mass of the gas molecule. Equation (1) is obtained based on the assumption of complete diffuse reflections after gas-wall collisions. The transmission probability, $\alpha_K$, introduced by Clausing[14], characterizes the efficiency of gas transport through nanochannels. That is, if the gas flow rate arriving at the channel entrance is $Q^{in}$, the flow rate leaving the exit is $\alpha_K Q^{in}$ while $(1 - \alpha_K)Q^{in}$ goes back to the gas reservoir through the entrance.

The diffuse reflection assumption does not hold under all circumstances. Experiments have confirmed the presence of specular reflections after gas molecules colliding with the wall[19–21]. Equation (1) could significantly underestimate the gas flow rate for some smooth surfaces[22–27]. Maxwell introduced the tangential momentum accommodation coefficient (TMAC), represented by $f$, and postulated that a fraction $f$ of the incident molecules will be reflected diffusely while the rest $1 - f$ will experience the specular reflection[28]. For the solely diffuse scattering, $f = 1$ and the complete specular reflection corresponds to

[1]CAS Key Laboratory of Mechanical Behavior and Design of Materials, Department of Modern Mechanics, University of Science and Technology of China, Hefei 230027, China. [2]State Key Laboratory of Nonlinear Mechanics, Institute of Mechanics, Chinese Academy of Science, 15 Beisihuan West Road, Beijing 100190, China. ✉e-mail: wuha@ustc.edu.cn; wangfc@ustc.edu.cn

$f = 0$, as illustrated in Fig. 1a. The TMAC has been involved to calculate the slip velocity at the gas-solid interface[8,21]. Smoluchowski[12] suggested a correction factor to the Knudsen theory and put forward $Q^S = \frac{2-f}{f} Q^K$. This theoretical amendment has been adopted to consider the flow enhancement caused by the specular reflection[24,29–31]. For recent experiments based on the fabricated nanodevices using materials with atomically smooth surfaces[24], the specular reflection could be dominant and the gas flow rate is extraordinarily higher than the prediction from Eq. (1). Moreover, the permeability of helium shows independent of the length of graphene channels, which suggests the ballistic molecular transport[24]. All these results implies that the influence of smooth surface on gas transport should be handled discreetly and the theoretical correction on $f$ is indispensable. However, there is an apparent imperfection that Smoluchowski model cannot work over a full range of $f$ from 0 to 1. When $f$ approaches to 0, $Q^S$ would diverge to infinity. Nonetheless, the outlet flow rate should be limited by the inlet flow rate through an aperture, which is the case of $\alpha_K = 1$ in Eq. (1). The ballistic transport of helium through graphene and hBN nanochannels can be analogy with ballistic electron transport in metallic systems and

semiconductors[32,33]. The entry resistance of gas transport is analogous to Sharvin resistance[34]. There are some other circumstances in which the specular reflection is dominant. For instance, the transport of electrons in quantum wires was analyzed and the results show that the probability of diffuse scattering, $f$, is close to 0[35,36]. Despite various discussions and revisions on Knudsen theory throughout the century[37–40], this issue remained unresolved.

In this work, we aim to develop a modified model to extend the application scope of Knudsen theory. The proposed model is capable to provide the theoretical estimation on the flow rate through nanochannels with various boundary conditions, especially for nanochannels with atomically smooth surfaces where the specular-reflection boundary condition should be applied. The validity of our proposed model was verified using molecular dynamics simulations, considering different channel shape, geometry size and surface roughness.

## Results

Let's start from briefly reviewing how Smoluchowski model was derived and then point out why it gives an inappropriate estimation on

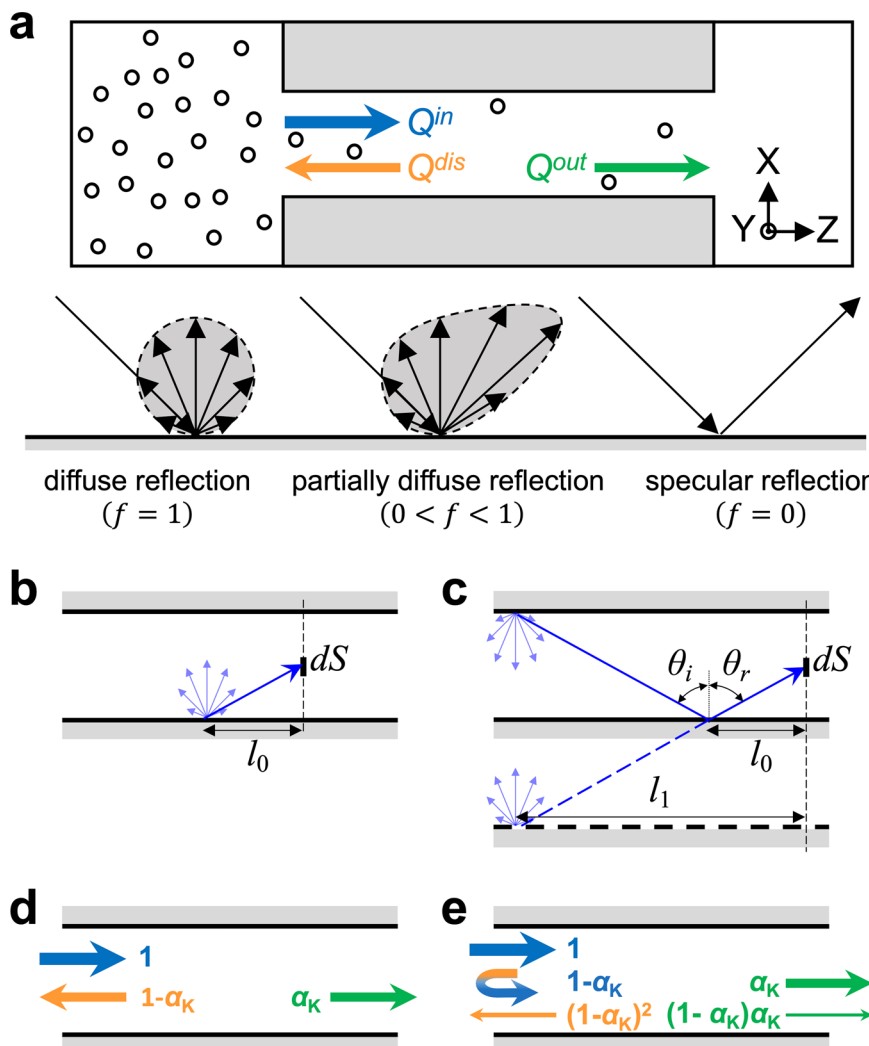

**Fig. 1 | Gas transport through nanochannels and theoretical interpretations on the flow enhancement due to specular reflection. a** Schematic of the gas transport through a nanochannel. Different boundary conditions of the free molecular flow are illustrated, including diffuse reflection, partially diffuse reflection, and specular reflection. **b** Illustration of the process of a gas molecule reaching the cross-section surface element $dS$ after diffuse emission from the wall. **c** Trajectory of a gas molecule arriving $dS$ after a specular reflection, which can be considered as being emitted from a location further away. **d** The normalized gas flow rate for inlet, outlet, and dissipation under the diffuse reflection assumption. **e** Sketch of the normalized gas flow rate after one specular reflection, which shows the reduction of the dissipation flow rate.

the gas flow rate for quite small $f$. When a gas molecule collides with channel walls somewhere and experiences the diffuse reflection, it can be equivalently treated as being emitted from the same point[12]. The emission direction follows the Lambert cosine law[41]. The gas flow rate can be obtained by integrating the emission rate through a surface element $dS$ over the entire cross-section, as illustrated in Fig. 1b. If the gas molecule undergoes one specular reflection before it arrives at $dS$, it can be regarded as being emitted from a location further away. For a specular reflection, the incidence angle equals to the reflection angle, which can help us tracing to the preceding diffuse reflection, as shown in Fig. 1c. The emission rate is proportional to the tangential distance between the emission point and $dS$[12]. In Fig. 1b, this distance is $l_0$ while it increases to $l_1$ after one specular reflection, as shown in Fig. 1c. Thus, we can expect that this specular reflection leads to an increase in the flow rate. According to Smoluchowski, one specular reflection results in a three-fold enhancement of the flow rate statistically, while $i$ times of specular reflections will lead to an enhancement of $2i + 1$ times[12]. For a certain solid surface that the probability of diffuse reflections is $f$, $i$ times of specular reflections occur with a probability of $f(1 - f)^i$. Accordingly, Smoluchowski model can be derived and the enhanced flow rate is written as,

$$Q^S = \sum_{i=0}^{\infty} f(1-f)^i (2i+1) Q^K = \frac{2-f}{f} Q^K. \tag{2}$$

According to the foregoing analysis, if $2i + 1$ is sufficiently large, the tangential emission distance $l_i$ would inevitably exceed the actual channel length $L$. Then the enhancement would be overestimated, which leads to the unphysical results.

To correct this issue, we consider the gas transport through nanochannels from a different perspective. For convenience, a vacuum side was set and $P^{out} = 0$. The molar flow rate through the inlet of the channel inlet is $Q^{in} = P^{in}A/\sqrt{2\pi RTM}$, while the molar flow rate from the channel outlet is $Q^{out}$. Because of the random diffuse scattering, some of gas molecules would eventually go back to the feed container, which is referred to as the dissipation flow rate in this work, $Q^{dis}$, as shown in Fig. 1a. Apparently, $Q^{in} = Q^{out} + Q^{dis}$. Under the complete diffuse-scattering boundary condition ($f = 1$), Eq. (1) gives $Q_0^{out} = Q^K = \alpha_K Q^{in}$. Therefore, $Q_0^{dis} = (1 - \alpha_K) Q^{in}$. The dissipation factor here is $\delta_0 = 1 - \alpha_K$, in which the subscript indicates how many times of specular reflection occur. This situation is illustrated in Fig. 1d. For a surface with $0 < f < 1$, we first consider that all gas molecules experience the specular reflection one time, and the corresponding probability is $p_1 = f(1 - f)^1$. After a specular reflection, the forward velocity component of each gas molecule can be maintained, which means that the specular reflection can enhance the outlet flow rate and suppress the dissipation flow rate. Compared with the complete diffuse-scattering case, we can image that $Q_0^{out}$ could still pass through the channel, yet some of $Q_0^{dis}$ would now contribute to the outlet flow rate. To a first approximation, $\alpha_K$, is inversely proportional to the channel length[6], $L$, indicating the dissipation flow rate can be reduced with the decreasing channel length. In other words, the consequence of specular reflection can be analogous to the reduced gas dissipation in a shorter channel, albeit with the complete diffuse-scattering boundary condition. This length variation can be estimated to be $\Delta L \propto H \tan \theta$, in which $H$ is the channel height and $\theta$ is the angle of this specular reflection. Unfortunately, it is rather complicated or impossible to calculate $\Delta L$ because $\theta$ cannot be precisely known for every reflection. Here, we consider that the specular reflection would equivalently result in one more diffuse reflection near the inlet. Because of this specular reflection, these gas molecules with a flow rate of $Q_0^{dis}$ behave as if they get another chance to re-enter the nanochannel. Then they travel to under the complete diffuse-scattering boundary condition, which can be quantitatively described by Eq. (1). Thus, the increment to the outlet gas flow rate is

$\triangle Q_1^{out} = \alpha_K Q_0^{dis}$. In consequence,

$$Q_1^{dis} = Q_0^{dis} - \alpha_K Q_0^{dis} = (1 - \alpha_K)^2 Q^{in}. \tag{3}$$

Now the dissipation factor is reduced to $\delta_1 = (1 - \alpha_K)^2$, as shown in Fig. 1e. Following this line of thought, gas molecule experience $i$ times of specular reflection in the nanochannel, with a probability of $p_i = f(1 - f)^i$, the dissipation factor becomes $\delta_i = (1 - \alpha_K)^{i+1}$. Combining $\delta_i$ and $p_i$, we can obtain the outlet flow rate,

$$Q^{out} = \sum_{i=0}^{\infty} p_i (1 - \delta_i) Q^{in} = \frac{\alpha_K}{f + \alpha_K - f\alpha_K} Q^{in}. \tag{4}$$

It is invigorating to find that Eq. (4) also satisfies two extreme cases. If $f = 1$, Eq. (4) becomes the original form of Knudsen theory, $Q^{out} = \alpha_K Q^{in}$. If $f = 0$, $Q^{out} = Q^{in}$, corresponding to the situation that the channel surface is extremely smooth. Equation (4) is what we are striving in the present work. This is a generalized Knudsen theory for gas transport through nanochannels under various kinds of boundary conditions of $0 \le f \le 1$. Moreover, Eq. (4) solves the inherent problem of Smoluchowski model, which is not applicable for $f = 0$.

From Eqs. (2) and (4), we can find that both analytical models express in the similar form of $\sum_{i=0}^{\infty} p_i q_i$, where $q_i$ is the enhanced gas flow rate due to $i$ times of specular reflections. They include the contribution from all possible specular reflections, together with the corresponding probability. Yet the underlying mechanism of these two theories is quite different. From Eq. (2), $q_i = (2i+1)\alpha_K Q^{in}$. Smoluchowski model focuses on the enhancement of outlet flow rate by the specular reflection. While in Eq. (4), $q_i = \left(1 - (1 - \alpha_K)^{i+1}\right) Q^{in}$. Our proposed model emphasizes that the specular reflection can lead to the reduced dissipation gas flow rate. To intuitively differentiate between these two models, we plotted the ratio of $q_i$ to $Q^{in}$ as a function of $i$ in Fig. 2a. The transmission probability $\alpha_K$ is dependent on the geometry of nanochannels, such as length, height, or radius[6,42-44]. It is a constant for the determined system. From Fig. 2a, we can find that $(2i+1)\alpha_K$ exceed 1 as $i$ increases, which is unrealistically. In contrast, $q_i/Q^{in}$ would eventually converge to 1 in our proposed model. We also learn from Eq. (2) that Smoluchowski model gives the enhancement factor is $\frac{2-f}{f}$. From this point of view, Eq. (4) can be rewritten as,

$$Q^{out} = \frac{1}{f + \alpha_K - f\alpha_K} Q^K. \tag{5}$$

Here the prefactor $1/(f + \alpha_K - f\alpha_K)$ is the enhancement factor compared with the gas flow rate predicted from Knudsen theory. The dependence of these two enhancement factors on $f$ are provided in Fig. 2b. Our proposed enhancement factor incorporates an additional parameter $\alpha_K$. Because $f - f\alpha_K > 0$, $1/(f + \alpha_K - f\alpha_K)$, which guarantees that $Q^{out}$ would not exceed the inlet flow rate $Q^{in} = Q^K/\alpha_K$, i.e., the enhancement factor is always smaller than $1/\alpha_K$, as shown in the inset to Fig. 2b. On the other hand, we can find from Fig. 2b that $1/(f + \alpha_K - f\alpha_K)$ has the other extreme value of 1 when $f = 1$, which recovers the Knudsen diffusion.

To evaluate the validity of our proposed model, molecular dynamics (MD) simulations were conducted to calculate the gas flow rate through nanochannels. Here we first investigated the transport of argon through a slit with a height of 6 nm and a length of 100 nm (see Methods and Supplementary Fig. 1). In MD simulations, $f$ can be obtained by calculating the TMAC[25] (Supplementary Fig. 2). We altered the surface roughness by changing the positions of the wall atoms and tuning the gas-wall interactions[45,46] (see Methods, Supplementary Table 1 and Supplementary Fig. 3). MD results of the gas flow rate as a function of $f$ were plotted in Fig. 3a. It is not easy in full-atom MD simulations to construct a nanochannel with a precise $f$, particularly for

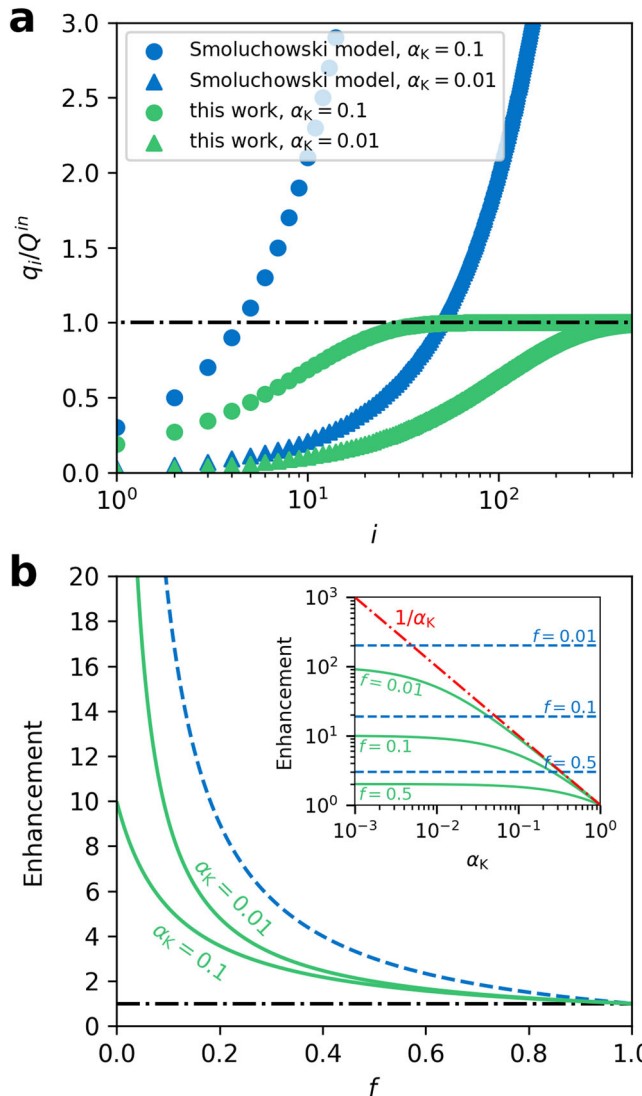

**Fig. 2 | Enhanced gas flow rate due to specular reflection. a** Enhanced gas flow rate, $q_i$, due to $i$ times of specular reflection, in relation to inlet flow rate, $Q^{in}$, at different $\alpha_K$. **b** The enhancement factor (ratio of outlet flow rate to that predicted by Knudsen theory) as a function of the fraction of diffuse reflection, $f$. The green solid lines were obtained using our model. The blue dashed lines were given by Smoluchowski model.

$f$ close to 0. To address this, we implemented virtual wall simulations in which an arbitrary value of $f$ between 0 and 1 can be assigned to the wall (see Methods). Then the outlet gas flow rate through the nanochannel with the same geometry were calculated over an extended range of $f$ and the results were also summarized in Fig. 3a. Compared to Smoluchowski model, Eq. (4) demonstrates a much better agreement with MD results, even when $f$ approaches 0. The predictions from Smoluchowski model diverges at small $f$, significantly overestimating the flow rate. The results obtained from virtual wall simulations are in quite good agreement with those from full-atom MD simulations, for channels with different surface roughness, height and length, as shown in Fig. 3b. We also employed our model to interpret the experiments of gas flow through carbon nanotube (CNT) membranes, as well as graphite and $MoS_2$ nanochannels, as shown in Fig. 3c. For CNT membranes[23], $f$ can be estimated to be in a range from 0.01 to 0.03, which is consistent with the intrinsic smoothness of the nanotube inner surface. Our model estimates $f \approx 0.001$ for most experimental results of graphite channels. Several exceptions, like $f \approx 0.1$ and $f \approx 0.3$ for $H \sim 4$–$8$ nm, can be understood by the hydrocarbon contamination

which is responsible for the changeover from ballistic to diffusive transport[24]. Using the experiments data of $MoS_2$ nanochannels, the estimated $f$ shows significant fluctuations, yet around 1.0. In addition, we analyzed our previous MD results[26], which yields $f \approx 0.1$ for the modeled graphene surfaces. We also plotted the enhancement factor, $Q^{out}/Q^K$ as a function of $L/H$ for different $f$ in Fig. 4d. It is quite reassuring to find that the generalized Knudsen theory shows good agreement with the numerical solution of Boltzmann equation[47], see Fig. 11 there. The proposed model provides a much simpler way to predict the gas flow rate through nanochannels, while numerically solving the Boltzmann equation is a challenge for experimentalists without specialized knowledge. All these results in Fig. 3 verify our proposed model from various perspectives.

For gas flow through channels with finite length, the end effect must be taken into account[10]. Pressure loss at the inlet and outlet of a finite channel can be identified in numerical solutions of Boltzmann kinetic equation[47] as well as our MD simulations (Supplementary Figure 4). The concept of effective channel length was proposed to assess influence of end effects on rarefied gas flows[47–49]. Under the diffuse reflection assumption, the evaluation of gas flow in the framework of Knudsen theory can be reduced to calculating $\alpha_K$, which is independent on gas pressure[6]. The Clausing-type integral equations and its series expansions can give $\alpha_K$ which is capable to describe gas flow rate for either long or short channels[10], indicating that the end effect has been integrated into $\alpha_K$. The derivation of Smoluchowski model neglects the fact that the channel has a finite length[11]. In this work, the suppressed dissipation flow rate due to the presence of specular reflections is evaluated under the complete diffuse-scattering boundary condition, as seen in Eq. (3). In this sense, the proposed model has taken account of the end effect.

We further tested the applicability of our proposed model in other scenarios, encompassing various shapes of the channel cross-section, variations in the geometry size over a wide range, and different boundary conditions. We performed virtual wall simulations to calculate the gas flow rate of slit channels with length-to-height ratios $L/H$ ranging from $10^{-1}$ to $10^4$. For intuitive comparison, the outlet-to-inlet flow rate ratios as a function of $L/H$ for $f$ which equals to 1, 0.7, and 0.2, respectively, were provided in Fig. 4a. These values of $f$ roughly correspond to surfaces of most common materials[50], some polished metal walls[51], and atomically smooth materials[25], respectively. For $f = 0.7$, Smoluchowski model provides satisfactory predictions when $L/H$ is large. However, for $f = 0.2$, it works well only when $L/H > 10^3$. We also investigated gas transport through nanochannels with other cross-sectional shapes, such as circle, rectangle and equilateral triangle. Comparison between theoretical and computational gas flow rate through channels with circular cross-section were summarized in Fig. 4b, which further demonstrates the applicability of our model. Nanochannels with a quite small length-to-diameter ratio (e.g., $L/D = 0.1$) can be considered as an aperture or a nanopore on a thin membrane[52]. There is no remarkable dependence of the flow rate ratio for quite small $L/D$ on $f$ (Supplementary Fig. 5), which again demonstrates that the proposed model manifests the entry resistance appropriately. For such an aperture, most gas molecules traverse the membrane without undergoing any gas-wall collisions (Supplementary Fig. 5). We also performed simulations of free molecule flow through channels with rectangular and equilateral triangular cross-sections, and again our model yields good estimations in these cases, as shown in Fig. 4c, d. For equilateral triangular cross-sectional nanochannels, the distance from the center of the triangle to the vertex is represented by $A$ and the $L/A$ ratio varies from 0.1 to 100. Our proposed model provides reasonable predictions for all these simulation results. Only when the channel's lateral size is comparable with the diameter of gas molecules, the generalized Knudsen theory should take into account the entry effect. The effective size at the channel entry also depends on the size and incident angle of gas molecules[24].

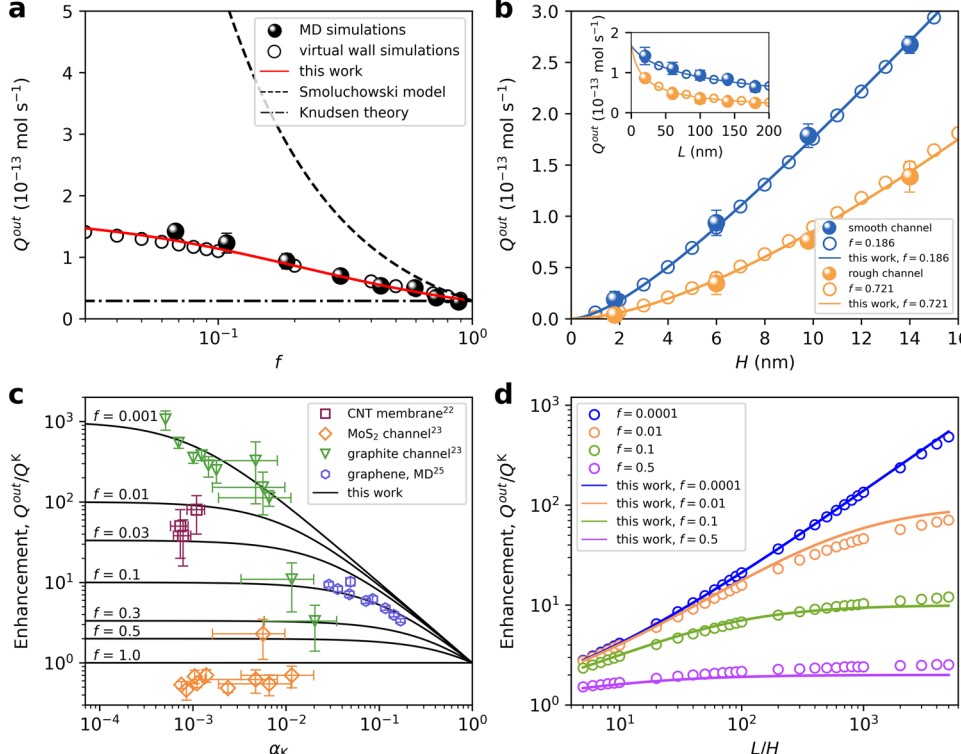

**Fig. 3 | Comparison of the generalized Knudsen theory with MD simulations, experiments and Boltzmann equation. a** Outlet flow rate based on full-atom MD simulations (solid circle symbols) and virtual wall simulations (open circle symbols) for slits with 6 nm in height and 100 nm in length, as a function of $f$. The dash-dotted, dashed, and solid lines represent theoretical results from Knudsen, Smoluchowski and our developed model, respectively. **b** The outlet flow rate for slits with different surface roughness, height (with a fixed length of 100 nm) and length (with a fixed height of 6 nm). Results from full-atom MD simulations (solid circle symbols) and virtual wall simulations (open circle symbols) are presented and

in good agreement. **c** Comparison of our proposed model with results from experiments and our previous MD simulations. **d** The enhancement factor as a function $L/H$ for slit channels, for different $f$, which show a remarkable consistency with the numerical solutions of Boltzmann equation[47]. Open circle symbols represent results from virtual wall simulations, and solid lines show our proposed model. Error bars in (**a**, **b**) depict the standard deviation derived from three molecular dynamics simulations. Error bars in **c** are based on data extracted from corresponding literature.

## Discussion

In summary, a generalized Knudsen theory was proposed to predict the gas flow rate though nanochannels. The classical Knudsen theory is valid only under diffuse reflection assumption and Smoluchowski model could not provide accurate estimation when the specular reflection is dominant. The applicability of the proposed model has been extended to all boundary conditions with the fraction of diffuse reflection, $f$, ranging from 0 to 1. The validity of our proposed model is supported by MD simulations on gas transport through nanochannels with diverse cross-sectional shapes and geometry sizes over a wide range. The algorithm virtual wall simulations suggest an efficient method to investigate the free molecular flow, in which an arbitrary value of $f$ between 0 and 1 can be set as the boundary condition. The generalized Knudsen theory is employed to interpret the recent experiments of gas flow through nanochannels. These findings offer a fundamentally new perspective for gas permeation through nanoscale pores, which plays an important role in many natural processes and modern technologies.

## Methods

### Methodology of molecular dynamics simulations

To calculate the gas flow rate, we constructed a nanochannel that is connected to a gas-filled feed side and a vacuum side (Supplementary Fig. 1). The temperature and pressure in the feed side are 300 K and 0.1 bar, respectively. The gas pressure in each side was controlled via adding new gas molecules into the feed side and deleting molecules arriving at the vacuum side. The gas flow rate was calculated by the derivative with respect to time of the number of deleted gas molecules.

The channel height $H$ available for gas transport was determined as $H_0 - t$, in which $H$ is the center-to-center distance of nearest atoms from two opposite walls and $t = 0.35$ nm is the effective thickness of one atomic layer (Supplementary Fig. 1). When $H$ is comparable with the diameter of gas molecules, the effective height at the channel entry also depends on the size and incident angle of gas molecules[24]. The height of the slit-channel is in the range from 2 nm to 20 nm. Argon was chosen to model the transport gas and its van der Waals diameter is about 0.376 nm[53]. The Knudsen number was estimated to be 33 to 330, which indicates that the studied flow is a typical free molecular flow[8]. The interactions between atom pairs within a cutoff distance 12 Å were calculated by the Lennard-Jones (LJ) potential,

$$V(r) = 4\varepsilon \left[ \left( \frac{\sigma}{r} \right)^{12} - \left( \frac{\sigma}{r} \right)^{6} \right], \quad (6)$$

where $r$ is the distance between interacting atoms, $\varepsilon$ determines the interaction strength, and $\sigma$ characterizes the effective atomic diameter. The adopted LJ parameters for argon atoms are $\varepsilon = 10.14$ meV and $\sigma = 0.34$ nm[54], and the parameters for channel atoms were artificially set as $m = 12.011$ u, $\varepsilon = 0.87$, 4.34, 8.67 or 17.3 meV and $\sigma = 0.35$, 0.40 or 0.50 nm. The Lorentz-Berthelot mixing rule was used to determine the LJ parameters between atoms of different types. The nanochannel has face-centered cubic (FCC) lattice with lattice constant equal to 3.5 Å. We investigated the influence of different $\varepsilon$ and $\sigma$ on $f$[45,46] (Supplementary Table 1 and Supplementary Fig. 3). We also shifted the position of surface atoms to obtain different surface roughness

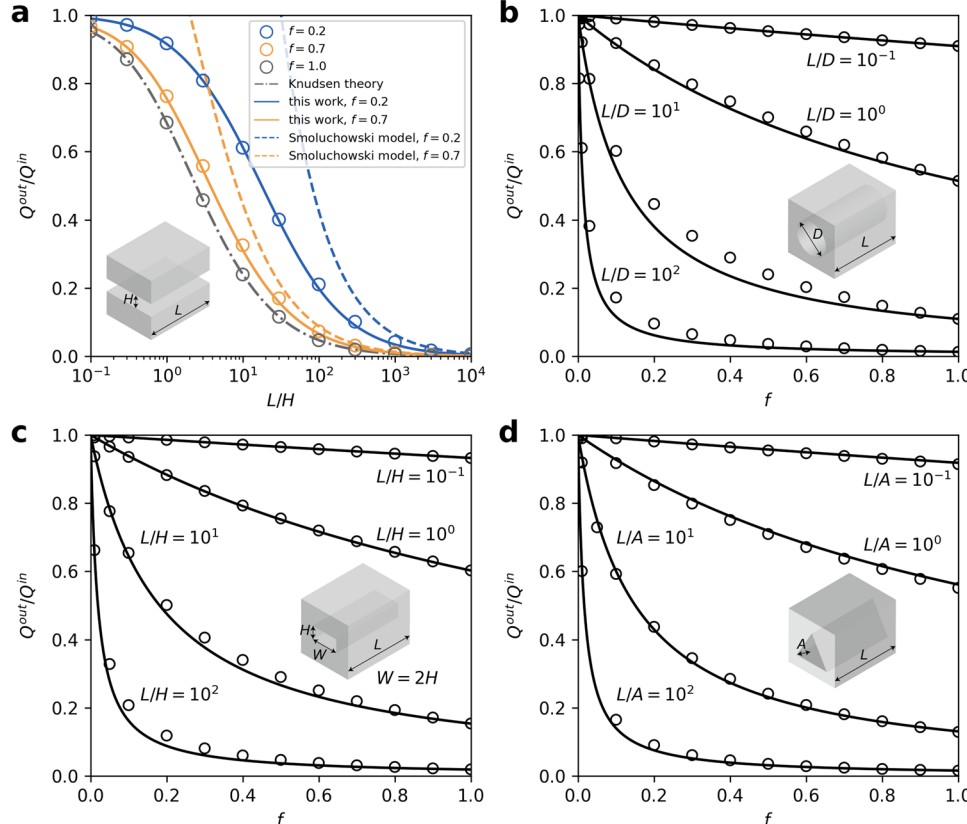

**Fig. 4 | Applicability of the proposed model in other scenarios. a** Slits with varying length-to-height ratios ($L/H$) over several orders of magnitude. **b**–**d** Theoretical and computational gas flow rate through nanochannels with (**b**) circular, (**c**) rectangular, and (**d**) equilateral triangular cross-section as a function of $f$, for the corresponding aspect ratios spanning multiple orders of magnitude. The open circle symbols represent the results of virtual wall simulations. The solid black lines denote the theoretical predictions of our proposed model. The insets illustrate the channel schematic.

(Supplementary Fig. 1). Specifically, each surface atom, initially at a position ($x_0$, $y_0$, $z_0$), is relocated to a new position ($x_0+dx$, $y_0+dy$, $z_0+dz$). Here, $dx$, $dy$ and $dz$ are random values generated in a range of [$-d_{max}$, $+d_{max}$]. Then the corresponding $f$ for different surfaces was calculated, listed in Supplementary Table 1. All the wall atoms were fixed during MD simulations. Periodic boundary conditions were implemented along both X and Y axes. The velocity-Verlet integrator was used to calculate trajectories of atoms with a timestep of 1 fs. The Berendsen thermostat was employed to modulate the temperature of gas[55]. MD simulations were performed using LAMMPS[56].

## Methodology of virtual wall simulations

Instead of using LJ particles to construct the channel walls, we defined an artificial surface with certain geometry as the virtual wall. Then the expected $f$ was assigned. A gas atom was located at the entrance of the channel with an initial velocity. When this gas atom 'collides' with the virtual wall, a random number between 0 and 1 is generated. If the random number is smaller than the assigned $f$, the particle will be assigned a random velocity. The direction of this velocity follows the cosine law[41], referred to as diffuse reflection. Otherwise, the velocity in the tangential direction remains unchanged, but its normal velocity reverses, known as specular reflection (Supplementary Fig. 6). Thus, we can model a nanochannel with a precise $f$, especially for $f$ close to 0. Eventually, the gas atom will either reach the exit, or return to the entrance. Such simulation was performed for $N = 10^6$ times. The initial position of gas in the entrance plane was randomly placed and the angle between its initial velocity and the channel central axis, $\varphi$, obeys the cosine law[6]. We counted how many times that the gas atom reached the exit, denoted as $N^{out}$. The ratio of the outlet flow rate to the inlet flow rate, $Q^{out}/Q^{in}$, can be calculated as $N^{out}/N$. Virtual wall

simulations were performed to calculate $Q^{out}/Q^{in}$ for channels characterized by different cross-section shape and different length. We found that in the case of $f = 1$, the calculated $Q^{out}/Q^{in}$ in virtual wall simulations is well in line with the results expected from the Knudsen theory[6,42–44] (Supplementary Fig. 7). When $f$ is set to be 0, the simulation results yield $Q^{out}/Q^{in} = 1$. To evaluate $Q^{out}$ in virtual wall simulations (Fig. 3a, b), we initially compute $Q^{in} = P^{in}A/\sqrt{2\pi RTM}$, assuming identical pressure, temperature, and molar mass in MD simulations. Then $Q^{out}$ was determined by the product of $Q^{in}$ and $Q^{out}/Q^{in}$.

## Calculation of the transmission probabilities

In simulations, the slit channel is a narrow gap formed between two infinitely wide slabs parallel to each other, each having a finite length, $L$. These slabs are separated by a distance, which corresponds to the height of the slit, $H$. Berman provided the transmission probability of a slit, suitable for any slit length[6,42],

$$\alpha_K^{slit} = 0.5\left(1+\sqrt{1+\chi^2}-\chi\right) - \frac{1.5\left[\chi - \ln\left(\chi + \sqrt{1+\chi^2}\right)\right]^2}{\chi^3 + 3\chi^2 + 4 - (\chi^2 + 4)\sqrt{1+\chi^2}}, \quad (7)$$

where $\chi$ is the ratio of $L$ to $H$. Berman additionally derived the transmission probability formula for a channel with a circular cross-section, with a length $L$ and a cross-sectional diameter $D$. The formula is given by[6,42],

$$\alpha_K^{circ} = 1 + \frac{\chi^2}{4} - \frac{\chi\sqrt{\chi^2+4}}{4} - \frac{\left[(8-\chi^2)\sqrt{\chi^2+4}+\chi^3-16\right]^2}{72\chi\sqrt{\chi^2+4}-288\ln\left(\chi+\sqrt{\chi^2+4}\right)+288\ln2}, \quad (8)$$

where $\chi$ represents the ratio of $L$ to $D/2$. When $\chi$ approaches infinity, Eq. (8) reduces to the classical expression introduced by Knudsen[11],

$$\alpha_K^{circ} = \frac{4D}{3L}.$$ (9)

For channels with a rectangular cross-section, Smoluchowski proposed an expression of the transmission probability for long channels[6,12],

$$\alpha_K^{rect} = \frac{H}{L}\left[\frac{\ln\left(\chi + \sqrt{1+\chi^2}\right)}{\chi} + \ln\left(\frac{1 + \sqrt{1+\chi^2}}{\chi}\right) + \frac{1 + \chi^3 - (1+\chi^2)^{3/2}}{3\chi^2}\right],$$ (10)

where $L$ is the channel length, and $\chi$ is the ratio of the channel's cross-sectional height $H$ to its width $W$, with $W > H$. Equation (10) is only applicable to long channels, rather than short ones. We utilize the $\alpha_K^{rect}$ values from Ref. 43, which were obtained via Monte Carlo calculations and cover a wide range of lengths and aspect ratios.

For long channel with an equilateral triangular cross-section[6],

$$\alpha_K^{tri} = \frac{3\ln 3}{2}\frac{A}{L},$$ (11)

where $A$ is the triangle's centroid-to-vertex distance, and $L$ is the channel length. In this work, $\alpha_K^{tri}$ comes from Monte Carlo simulations[44], taking account of both short and long channels.

## Data availability
Source data are provided with this paper, which are available on Figshare (https://doi.org/10.6084/m9.figshare.24218886).

## Code availability
The python code for the virtual wall simulations is available on Figshare (https://doi.org/10.6084/m9.figshare.24218886)

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

## Acknowledgements

This work was financially supported by the National Natural Science Foundation of China (12241203, U22B2075, F.C.W.) and the Youth Innovation Promotion Association CAS (2020449, F.C.W.). The numerical calculations were performed on the supercomputing system in Hefei Advanced Computing Center and the Supercomputing Center of University of Science and Technology of China.

## Author contributions

F.C.W. conceived the project, J.H.Q. performed the research, J.H.Q., H.A.W., and F.C.W. discussed and wrote the manuscript.

## Competing interests

The authors declare no competing interests.
