## [Peer Review File · Nature Communications]

REVIEWER COMMENTS

Reviewer #1 (Remarks to the Author):

In this paper, the authors incorporated the effects of specular scattering of gas molecules at solid surfaces into the Knudsen theory and derived a more general formula to predict the mass flow rate of a pressure-driven free molecular flow in a channel. Both molecular dynamics simulations and Monte Carlo simulations were carried out to validate the theoretical prediction. The modeling results show that the correction factor derived by the authors gives consistently reasonable prediction of mass flow rate of free molecular flows as the fraction of diffuse scattering (f) varies from 0 to 1. The authors show that the new formula is more accurate than the Knudsen theory and the Smoluchowski correction to the Knudsen theory as the surface scattering becomes more specular.

The derivation and validation of the correction to the Knudsen theory seem reasonable to me, but I do not believe that the physical understanding of the molecular flow provided in this paper is novel enough to warrant publication in Nature Communications.

(1) The significant enhancement of mass flow rate of free molecular flow by specular surface scattering has been observed by both experiments and molecular dynamics simulations. The Smoluchowski model was shown to give reasonable understanding of gas transport phenomena in experiment. For example, using the Smoluchowski model, the experimental work in Ref. [23] found $f = 0.2$ for a monatomic gas on a graphite nanochannel. This value has a reasonable agreement with the TMAC found in the MD simulation in Ref. [25]. If the authors use their improved model to analyze the experimental data in Ref. [23], will they get a significantly different f value compared to that predicted by the Smoluchowski model? Is the f value predicted by the new model consistent with the TMAC determined from the MD simulation?

(2) In the last line of page 2, "When f approaches to 0, Q_S would diverge to infinity." When f approaches zero, the flow resistance in the channel approaches zero. This means the flow conductance (i.e., the inverse of the flow resistance) in the channel diverges to infinity. In this case, the mass flow rate is not infinite because one should also consider the end effects on the flow resistance such as the flow resistance caused by the entry aperture. Does the improved model developed by the authors include the flow resistance at the ends of the nanochannel?

(3) In MD simulations, how did the authors determine the channel height h ? Determination of an accurate h value is important in theoretical prediction of Q .

(4) The authors should give more details of how the rough surfaces are generated. All solid atoms are fixed in the MD simulation. It is not clear how the authors generate a realistic rough surface in this case. To adjust the TMAC value, I think it is probably more realistic to vary the solid-gas interaction strength as the solid-gas interaction strength used in the MD simulation of this paper seems to be very low.

Minor points:

Eq. (1) and other similar equations in the paper give the molar flow rate, not molar flux. Q/A is the molar flux.

Near the bottom of page 2, k in Q_k is a subscript. But in other parts of the paper, k is a superscript. The authors should be consistent with the notation of Q_K .

Reviewer #2 (Remarks to the Author):

The authors present a generalization of the Knudsen theory and are able to remove the unphysical behaviour of the Smoluchowski model. The new equation for flow through nanochannels is then supported by MD simulations.

This paper is well written with a clear motivation.

However, it misses a comparison with experiment and therefore is of limited interest.

There are several experiments that could (or not) validate the new theoretical results. E.g. Ref. 23 present results for nanochannels made out of three different materials: MoS₂, Graphene and hBN.

There is also an earlier work on specular and diffusive transport of electrons in quantum wires which was also understood in terms of classical transport; see T. Thornton et al, PRL 63, 2128 (1989).

Reviewer #3 (Remarks to the Author):

The free-molecular flow has many important engineering applications, while the traditional analytical solution only works for special types of gas-surface interaction. The manuscript presents a theoretical derivation of the end effect (which only manifests when the ratio of the channel height to channel length is large) on the permeability over the whole range of the tangential momentum accommodation coefficient and typical cross-section shapes. I think the paper can be published in Nature Communications when the following problems are addressed:

(1) In figure 2 the statement that q_i/Q_{in} is somehow misleading. Please replace $PA/\sqrt{2\pi}RTM$ by other symbols, since this is not the inflow mass flow rate. Similarly, the statement in the line 51 that “the outlet flux could be probably even larger than the inlet flux” should be re-worded.

(2) The authors conducted the MD simulations, and showing only the results of flow rate. It will be useful to show also the velocity profiles as well as the pressure distribution along the channel. Especially the latter is quite important since from the authors' analysis in Fig 1e that each part of the channel contributed to the reduction of permeability when the channel is not long enough. However, it seem from the preprint in https://www.researchgate.net/publication/367021943_Highly_rarefied_gas_flows_in_rough_channels_of_finite_length that the end effect is only significant in the vicinity of inlet and outlet. If this is the case, the theoretical derivation, albeit giving accurate prediction of flow rate, is not physically justified.

(3) Further to the second point, I believe α_K takes into the end effect when the diffuse scattering is considered. Thus, it has little to do with the α_K in the intermediate region of the channel.

(4) Papers on the end effect should be mentioned to connected the community in rarefied gas dynamics and communities in MD simulations and reservoir engineering. Especially, I think the data in the presented paper is covered in the Figure 12 of the preprint mentioned in (2).

Point-by-Point Response to Reviewers' Comments

Response to comments of Reviewer #1:

In this paper, the authors incorporated the effects of specular scattering of gas molecules at solid surfaces into the Knudsen theory and derived a more general formula to predict the mass flow rate of a pressure-driven free molecular flow in a channel. Both molecular dynamics simulations and Monte Carlo simulations were carried out to validate the theoretical prediction. The modeling results show that the correction factor derived by the authors gives consistently reasonable prediction of mass flow rate of free molecular flows as the fraction of diffuse scattering (f) varies from 0 to 1. The authors show that the new formula is more accurate than the Knudsen theory and the Smoluchowski correction to the Knudsen theory as the surface scattering becomes more specular.

The derivation and validation of the correction to the Knudsen theory seem reasonable to me, but I do not believe that the physical understanding of the molecular flow provided in this paper is novel enough to warrant publication in Nature Communications.

Response: We would like to express our sincere gratitude to this Reviewer for the thorough review and insightful comments. In the revision, special attention has been paid to (i) the comparison between the proposed theory and the existing experiments, and (ii) the end effect, which raised most of the concerns from the Reviewers. We replotted Figure 3 and Figure 4 in the manuscript to provide more verifications of our proposed model from various perspectives. We also emphasized that this work offers a much simpler way to predict the gas flow rate through nanochannels, while numerically solving the Boltzmann equation is a challenge for experimentalist without specialized knowledge. Hope the revised version is now satisfying.

(1) The significant enhancement of mass flow rate of free molecular flow by specular surface scattering has been observed by both experiments and molecular dynamics simulations. The Smoluchowski model was shown to give reasonable understanding of gas transport phenomena in experiment. For example, using the Smoluchowski model, the experimental work in Ref. [23]

found $f = 0.2$ for a monatomic gas on a graphite nanochannel. This value has a reasonable agreement with the TMAC found in the MD simulation in Ref. [25]. If the authors use their improved model to analyze the experimental data in Ref. [23], will they get a significantly different f value compared to that predicted by the Smoluchowski model? Is the f value predicted by the new model consistent with the TMAC determined from the MD simulation?

Response: We appreciate very much the Reviewer for this comment. As a brief answer first, our model can provide better estimation on f than the Smoluchowski model and the results are consistent with the TMAC determined from MD simulations.

On the one hand, f can be obtained by calculating the TMAC in MD simulations. On the other hand, f is not easy to be measured directly from experiments. We can estimate f based on the flow enhancement compared with the results predicted from Knudsen theory. The Smoluchowski model gives an enhancement factor of $\frac{2-f}{f}$, while our model yields $\frac{1}{f+\alpha_K-f\alpha_K}$. The results are summarized in Fig. R1. We can find that the Smoluchowski model and our model show good agreement for large f (rough surfaces) and small α_K (long channels, or large L/H), as shown in Fig. R1(a). However, for small f (smooth surfaces) or large α_K (short channels, or small L/H), there are significant differences between the two models, as shown in Fig. R1(b & c). For the same enhancement factor, our model predicts a smaller f .

Fig. R1. The gas flow enhancement of as a function of f using our model (solid lines) and the Smoluchowski model (dashed lines) for different L/H . **a** $L/H = 1000.0$. **b** $L/H = 100.0$. **c** $L/H = 10.0$.

Fig. R2. Comparison of our proposed model with experiments and our previous MD simulations.

In our revised manuscript, we added Fig. 3c to compare our model with the experimental results and MD simulations. For convenience, we copied it here as Fig. R2. We can learn from this figure:

- (i) Using the experiments data of helium transport through 4-nm graphite channels [23], the Smoluchowski model gives $f \approx 0.2$. Our model yields $f \approx 0.1$ (the green triangle point marked by a red circle). In fact, MD simulations in Ref. [25] suggest that the TMAC for the helium/graphene system is about 0.1 (Fig. 4a in Ref. [25]). Using our model, MD simulations on the gas transport through graphene channels with different geometries can predict $f \approx 0.1$ (blue hexagonal points in Fig. R2).
- (ii) Using the enhancement factor obtained in experiments for graphite nanochannels [23], f for graphene surfaces can be estimated to be 0.001 (green triangle points in Fig. R2). However, we may find two green points at large α_k ($H \approx 4-8$ nm) correspond to $f \approx 0.1$ or $f \approx 0.3$. It is not surprising since the hydrocarbon contamination was suggested to explain the changeover from ballistic to diffusive transport in Ref. [23].
- (iii) Using the enhancement factor obtained in experiments for carbon nanotubes, f can be estimated to be about 0.01 to 0.03 (magenta square

points in Fig. R2), which is consistent with the intrinsic smoothness of the nanotube inner-surface [22].

In summary, we believe that our model can provide better estimation on f , especially for smooth surfaces and short channels.

(2) In the last line of page 2, “When f approaches to 0, Q^S would diverge to infinity.” When f approaches zero, the flow resistance in the channel approaches zero. This means the flow conductance (i.e., the inverse of the flow resistance) in the channel diverges to infinity. In this case, the mass flow rate is not infinite because one should also consider the end effects on the flow resistance such as the flow resistance caused by the entry aperture. Does the improved model developed by the authors include the flow resistance at the ends of the nanochannel?

Response: We appreciate this concern. We agree with the Reviewer that the flow rate should not be infinite due to the presence of the flow resistance at the channel end. However, the Smoluchowski model with an enhancement factor of $\frac{2-f}{f}$ would give unphysical results. In this work, we proposed the model

$$Q^{out} = \frac{1}{f + \alpha_K - f\alpha_K} Q^K = \frac{\alpha_K}{f + \alpha_K - f\alpha_K} \frac{\Delta PA}{\sqrt{2\pi RTM}}.$$

When f approaches zero, Q^{out} becomes $\frac{\Delta PA}{\sqrt{2\pi RTM}}$, which is equal to the flow rate through an aperture. Thus, our model appropriately includes the flow resistance at the ends of the nanochannel.

Dushman proposed an expression to channels with finite length. He postulated that the flow resistance of a short tube is the sum of the resistances of the entrance aperture and that of a long tube [12]. However, Clausing proved that the Dushman’s expression can only be regarded as a rough approximation. Instead, he introduced the concept of transmission probability, α_K , and derived an integral equation to calculate α_K [13]. Transmission probability can provide quite accurate results [9]. This integral equation as well as its series expansions for α_K work for both long and short channels. In this sense, α_K and its correction factor $\frac{1}{f + \alpha_K - f\alpha_K}$ have taken account of the end effect.

(3) In MD simulations, how did the authors determine the channel height h ? Determination of an accurate h value is important in theoretical prediction of Q .

Response: We thank the Reviewer for the insightful comment. In our MD simulations, the channel height H available for gas transport was determined as $H_0 - t$, in which H_0 is the center-to-center distance of nearest atoms from two opposite walls and $t = 3.5 \text{ \AA}$ is the effective thickness of one layer of solid atoms, as shown in Fig. R3a.

It should be noted that we also modified the force field parameters of the wall atoms or make symmetric and uniform random displacements to the surface atoms to achieve different roughness for the wall. We believe that these setting would not change the channel height significantly.

To further validate the suitability of our measured channel height in predicting flow rates, we have conducted additional simulations for channels with varying heights. It can be observed that the molecular dynamics simulation results are in good agreement with the theoretical values calculated based on our measured heights (Fig. R3b).

Fig. R3. a Illustration of determining the channel height for both smooth and rough channels. **b** The gas flow rates through smooth and rough channels with different heights. Solid circles indicate results obtained from full-atom MD simulations, while the lines represent predictions from the model proposed in this work. Error bars depict the standard deviation derived from three separate MD simulations.

Following the comment, we provide more details about our MD simulations in the corresponding section of Methods.

(4) The authors should give more details of how the rough surfaces are generated. All solid atoms are fixed in the MD simulation. It is not clear how the authors generate a realistic rough surface in this case. To adjust the TMAC value, I think it is probably more realistic to vary the solid-gas interaction strength as the solid-gas interaction strength used in the MD simulation of this paper seems to be very low.

Response: We thank the Reviewer for raising this point. This comment was quite helpful. In our study, the rough surfaces were modeled by changing the position or the size of the wall atoms. Specifically, each surface atom, initially at a position (x_0, y_0, z_0) , is relocated to a new position (x_0+dx, y_0+dy, z_0+dz) . Here, dx , dy , and dz are random values generated in a range of $[-d_{max}, +d_{max}]$. Then the corresponding f for different surfaces was calculated, listed in Supplementary Table 1.

We also modified the effect radius of wall atoms by adjusting the parameter σ in the Lennard-Jones (LJ) potential function. Increasing σ results in a smooth equipotential surface between the confining wall and gas molecules, leading to a reduction in TMAC. This effect has been discussed in our previous work [44].

Supplementary Table 1. Parameters for channels in MD simulations

Channel ID	σ (nm)	ϵ (meV)	d_{max} (nm)	TMAC, f
1	0.35	0.87	0	0.186
2	0.35	0.87	0.03	0.303
3	0.35	0.87	0.05	0.436
4	0.35	0.87	0.10	0.721
5	0.40	0.87	0	0.108
6	0.50	0.87	0	0.068
7	0.35	4.34	0	0.323
8	0.35	8.67	0	0.403
9	0.35	17.3	0	0.491
10	0.35	8.67	0.03	0.594
11	0.35	8.67	0.10	0.883

Following the Reviewer's suggestion, we also adjust the TMAC value by changing the solid-gas interaction strength. We changed the parameter ϵ in the LJ potential function for the surface atoms, ranging from 0.87 meV to 17.3 meV.

To offer a clear overview of the surface treatments, we've tabulated the different approaches and assigned IDs to each set of parameter combinations as listed in Supplementary Table 1. The corresponding f for different surfaces was calculated. We also plotted the outlet flow rate as a function of f , as shown in Fig. R4.

Both the table and Fig. R4 have been included in the supplementary information of the revised manuscript.

Fig. R4. (Supplementary Figure 3.) Argon gas flow rates through slit channels with different surface roughness quantified using f . The numbers beside the data points indicate the corresponding channel ID listed in Supplementary Table 1. The slit channel has a length of 100 nm and a height of 6 nm. Error bars depict the standard deviation derived from three molecular dynamics simulations.

Minor points:

Eq. (1) and other similar equations in the paper give the molar flow rate, not molar flux. Q/A is the molar flux.

Response: We are grateful for this comment and apologize for this carelessness. All the related expressions have been revised accordingly.

Near the bottom of page 2, k in Q_k is a subscript. But in other parts of the paper, k is a superscript. The authors should be consistent with the notation of Q_K .

Response: Let us thank the Reviewer again. This has been corrected in the revised manuscript.

Response to comments of Reviewer #2:

The authors present a generalization of the Knudsen theory and are able to remove the unphysical behaviour of the Smoluchowski model. The new equation for flow through nanochannels is then supported by MD simulations.

This paper is well written with a clear motivation.

Response: We are grateful for this kind assessment.

However, it misses a comparison with experiment and therefore is of limited interest.

There are several experiments that could (or not) validate the new theoretical results. E.g. Ref. 23 present results for nanochannels made out of three different materials: MoS₂, Graphene and hBN.

Response: We really appreciate this suggestion. In the revised manuscript, we added Fig. 3c in which we compared our theory with experiments, not only Ref. [23], but also Ref. [22]. The experiments can validate our proposed model.

Figure 3c in the revised manuscript.

We can learn from this figure:

(i) Using the enhancement factor obtained in experiments for carbon nanotube (CNT) membranes, f can be estimated to be about 0.01 to 0.03 (magenta square points in the Fig. 3c), which is consistent with the intrinsic smoothness of the CNT inner-surface [22].

(ii) Using the experiments data of helium transport through 4-nm graphite channels [23], the Smoluchowski model gives $f \approx 0.2$. Our model yields $f \approx 0.1$ (the green triangle point marked by a red circle). In fact, MD simulations in Ref. [25] suggest that the TMAC for the helium/graphene system is about 0.1 (Fig. 4a in Ref. [25]). Using our model, MD simulations on the gas transport through graphene channels with different geometry can predict $f \approx 0.1$ (blue hexagonal points in Fig. 3c).

(iii) Using the enhancement factor obtained in experiments for graphite channels [23], f for graphene surface can be estimated to be 0.001 (green triangle points in Fig. 3c). However, we may find two green points at large α_K ($h \approx 4\text{--}8$ nm) correspond to $f \approx 0.1$ or $f \approx 0.3$. It is not surprising since the hydrocarbon contamination was suggested to explain the changeover from ballistic to diffusive transport in Ref. [23].

(iv) We didn't compare with the experiments data for hBN nanochannels, because the experiments for hBN and graphite nanochannels reported in Ref. 23 exhibit quite close results.

(v) Using the experiments data of MoS₂ nanochannels, the estimated f shows significant fluctuations, yet around 1.0.

Moreover, our theory shows good agreement with the numerical solution of Boltzmann equation, see Fig. R7.

All the results provided in the revised Fig. 3 could verify our proposed model from various perspectives.

There is also an earlier work on specular and diffusive transport of electrons in quantum wires which was also understood in terms of classical transport; see T. Thornton et al, PRL 63, 2128 (1989).

Response: Many thanks on this comment. It is pleasant to learn that there are many similarities between the ballistic transport of helium through graphene nanochannels and the ballistic electron transport in metallic systems as well as semiconductors. We feel that the readers will be interested in this PRL paper and now cite it in the revised manuscript. We added the flowing sentences into Introduction.

Nonetheless, the outlet flow rate should be limited by the inlet flow rate through an aperture, which is the case of $\alpha_K = 1$ in Eq. (1). The ballistic transport of helium through graphene and hBN nanochannels can be analogy with ballistic electron transport in metallic systems and semiconductors^{31,32}. The entry resistance of gas transport is analogous to Sharvin resistance³³. There are some other circumstances in which the specular reflection is dominant. For instance, the transport of electrons in quantum wires was analyzed and the results show that the probability of diffuse scattering, f , is close to 0^{34,35}.

Response to comments of Reviewer #3:

The free-molecular flow has many important engineering applications, while the traditional analytical solution only works for special types of gas-surface interaction. The manuscript presents a theoretical derivation of the end effect (which only manifests when the ratio of the channel height to channel length is large) on the permeability over the whole range of the tangential momentum accommodation coefficient and typical cross-section shapes. I think the paper can be published in Nature Communications when the following problems are addressed:

Response: We thank the Reviewer for these kind words.

(1) In figure 2 the statement that q_i/Q_{in} is somehow misleading. Please replace $PA/\sqrt{2\pi RTM}$ by other symbols, since this is not the inflow mass flow rate. Similarly, the statement in the line 51 that “the outlet flux could be probably even larger than the inlet flux” should be re-worded.

Response: We appreciate these suggestions. We have revised P to P^{in} . The previous misunderstanding comes from the fact that P^{out} was set to zero in our MD simulations. Thus, $\Delta P = P^{in}$. We also modified Eq. (1) and the relevant description to clarify this.

The Knudsen theory gives the molar flow rate through a nanochannel connecting two gas reservoirs at pressure P^{in} and P^{out} , in a form of

$$Q^K = \alpha_K \frac{\Delta P A}{\sqrt{2\pi RTM}}, \quad (1)$$

in which $\Delta P = P^{in} - P^{out}$ is the pressure drop along the nanochannel.

As for the sentence, “For small f , the estimated outlet flux could be probably even larger than the inlet flux, which is physically impossible.” This is an inference from $Q^S = \frac{2-f}{f} Q^K$ for quite small f . We have pointed out that it is physically impossible. In our revision, this sentence has been removed and

replaced by “Nonetheless, the outlet flow rate should be limited by the inlet flow rate through an aperture, which is the case of $\alpha_K = 1$ in Eq. (1).”

(2) The authors conducted the MD simulations, and showing only the results of flow rate. It will be useful to show also the velocity profiles as well as the pressure distribution along the channel. Especially the latter is quite important since from the authors' analysis in Fig 1e that each part of the channel contributed to the reduction of permeability when the channel is not long enough. However, it seem from the preprint in [https://www.researchgate.net/publication/367021943 Highly rarefied gas flows in rough channels of finite length](https://www.researchgate.net/publication/367021943_Highly_rarefied_gas_flows_in_rough_channels_of_finite_length) that the end effect is only significant in the vicinity of inlet and outlet. If this is the case, the theoretical derivation, albeit giving accurate prediction of flow rate, is not physically justified.

Response: We thank the Reviewer for recommending this preprint, which is quite helpful. We certainly agree with the Reviewer that for gas flow through short channels, the end effect must be taken into account. As the Reviewer obviously understands, the gas transport in the free-molecular flow regime cannot be described by the Navier-Stokes equations. Instead, we have to turn to either the Boltzmann equation or the Knudsen diffusion theory.

However, we think these two theoretical approaches treat the end effect quite differently. When numerically solving the Boltzmann equation, the end effect is considered as the pressure loss at the inlet and outlet of a finite channel. For the Knudsen diffusion theory, the end effect is integrated into the transmission probabilities, α_K . The Clausing-type integral equations can give α_K which is capable to describe gas flow rate for channels with finite length. The transmission probabilities α_K can be evaluated using series expansions with high accuracy (less than 1%) [9]. Thus for $f = 1$ (solely diffuse scattering), $Q^{out} = \alpha_K Q^{in}$ can provide enough accurate results. Based on our understanding, the calculation on α_K only relies on the channel length and its cross-sectional shape, rather than the pressure distribution.

In the cases of smooth channels with quite small f , the end effect is significant (also verified in Figure 10 of the preprint). In this work, we focus on the

appropriate correction on f , especially when $0 < f < 1$. Previous theories suggested the correction factor $\frac{2-f}{f}$, which we believe is not applicable for quite small f .

In our work, the pressure in the gas-filled feed side was only used to estimate the molar flow rate through the inlet of the channel, $Q^{in} = \frac{P^{in}A}{\sqrt{2\pi RTM}}$. Even if there is a pressure loss at the inlet, the flow rate can be appropriately calculated. To verify this, we have carried out MD simulations on gas transport through thin orifices (the channel length $L \sim 0$). We expected that the molar flow rate through this orifice is $Q^{out} = \frac{PA}{\sqrt{2\pi RTM}}$, in which P is the gas pressure in the gas-filled feed side, as shown in Fig. R5a. The other side of the simulation box was maintained to be vacuum. There is a pressure loss near the orifice. The MD results for $P = 10^4$ Pa and $H = 2, 6, 10$ and 14 nm agree well with the theoretical prediction, as shown in Fig. R5b.

Fig. R5b. Argon gas flow through apertures. **a** Schematic of argon gas flow through a slit-like aperture with a height H . The black plates represent the wall, and the blue spheres depict argon atoms. **b** Flow rate of argon gas as a function of the height of the slit-like aperture. Black circles represent results from MD simulations. Error bars indicate the standard deviation from three distinct 2 ns MD simulations.

Following the Reviewer's comment, we discussed thoroughly with the recommended preprint in the revised manuscript. Moreover, we have calculated and provided the velocity profiles as well as the pressure distribution along the channel in the Supplementary Figure 4.

Fig. R6. (Supplementary Figure 4.) Pressure distribution along the channel and velocity profiles. **a** Normalized pressure distribution (relative to the feed side pressure of 100 mbar) of argon gas through the slit with smooth walls ($f = 0.186$). The black bold lines represent the channel walls. **b** Same as **a**, but for a rough-walled slit ($f = 0.721$). **c** Profiles of normalized gas pressure along the axial direction of the channel with smooth walls ($f = 0.186$). **d** Same as **c**, but for a rough-walled slit ($f = 0.721$). **e** Distribution of the average gas velocity of the argon flow in a slit with smooth walls ($f = 0.186$). **f** Same as **e**, but for a rough-walled slit ($f = 0.721$). **g** Profiles of gas average velocity at the channel entrance and mid-channel with smooth walls ($f = 0.186$). **h** Same as **g**, but for a rough-walled slit ($f = 0.721$). All the slit channels have a length of 20 nm and a height of 6 nm.

(3) Further to the second point, I believe α_K takes into the end effect when the diffuse scattering is considered. Thus, it has little to do with the α_K in the intermediate region of the channel.

Response : When P. Clausing introduced the concept of transmission probability, he derived an integral equation to calculate α_K [13]. This integral

equation works for both long and short channels. Thus, we agree here that α_K takes into account the end effect.

Once the channel geometry (channel length and cross-sectional shape) is determined, the value of α_K can be obtained. In this sense, α_K does not distinguish between the intermediate region of the channel and the entrance/exit of the channel.

The aforementioned discussion is based on the assumption of diffuse scattering, $f = 1$. For partial diffuse reflection, $0 < f < 1$, $Q^{out} = \frac{2-f}{f} \alpha_K Q^{in}$ is not accurate. We proposed in this work that $Q^{out} = \frac{1}{f + \alpha_K - f \alpha_K} \alpha_K Q^{in}$.

To address this comment, we have revised our manuscript and added more discussion on α_K .

(4) Papers on the end effect should be mentioned to connected the community in rarefied gas dynamics and communities in MD simulations and reservoir engineering. Especially, I think the data in the presented paper is covered in the Figure 12 of the preprint mentioned in (2).

Response: We thank the Reviewer for this helpful comment. In the revised manuscript, a paragraph was added to discuss the end effect.

For gas flow through channels with finite length, the end effect must be taken into account⁹. Pressure loss at the inlet and outlet of a finite channel can be identified in numerical solutions of Boltzmann kinetic equation⁴⁶ as well as our MD simulations (Supplementary Figure 4). The concept of effective channel length was proposed to assess influence of end effects on rarefied gas flows⁴⁶⁻⁴⁸. Under the diffuse reflection assumption, the evaluation of gas flow in the framework of Knudsen theory can be reduced to calculating α_K , which is independent on gas pressure⁵. The Clausing-type integral equations and its series expansions can give α_K which is capable to describe gas flow rate for either long or short channels⁹, indicating that the end effect has been integrated into α_K . The derivation of Smoluchowski model neglects the fact that the channel has a finite length¹¹. In this work, the suppressed dissipation flow due to the

presence of specular reflections is evaluated under the complete diffuse-scattering boundary condition, as seen in Eq. (3). In this sense, the proposed model has taken account of the end effect.

Following the Reviewer's suggestion, we plotted our results on flow enhancement coefficient as a function of L/H , as shown in Figure R7a, which are compared with Figure 11 of the preprint (Figure R7b). (We feel a bit puzzled by Figure 12 because there is no Figure 12 in the preprint, but eventually guessed that the Reviewer probably mean Figure 11.)

Fig. R7. a The enhancement as a function of L/H for different f . **b** Figure 11 of the preprint by Z. Shi, Y. Zhao, W. Su and L. Wu.

It is reassuring to find our modified model based on the Knudsen diffusion theory shows quite good agreement with the numerical solution of Boltzmann equation.

First, the excellent agreement between our results and Figure 11 in the preprint further verifies our proposed model from another perspective. Moreover, the end effects have been handled properly in the work of Z. Shi *et al*, which indicates that our proposed model includes the end effects appropriately.

Second, the flow enhancement factor K , (or the flow rate Q^{out}) due to the specular reflections on the smooth surface, can be estimated in a much simpler way, using the following analytical equations (taking the slit channel as an example):

$$K = \frac{Q^{out}}{Q^K} = \frac{1}{f + \alpha_K - f\alpha_K}$$

$$\alpha_K^{slit} = 0.5 \left(1 + \sqrt{1 + \chi^2} - \chi \right) - \frac{1.5 \left[\chi - \ln \left(\chi + \sqrt{1 + \chi^2} \right) \right]^2}{\chi^3 + 3\chi^2 + 4 - (\chi^2 + 4)\sqrt{1 + \chi^2}}$$

The ballistic molecular transport has been reported in the experiments, which urges for the development of theoretical tools to address the predominant specular reflections on atomically smooth surfaces. On the other hand, numerically solving the Boltzmann equation is a challenge for experimentalist without specialized knowledge.

Let us thank the Reviewer again for his/her thorough report.

REVIEWERS' COMMENTS

Reviewer #1 (Remarks to the Author):

I think the author addressed most of my concerns. But I still have one concern about the determination of channel height in their model. In the reply, the authors mentioned that they determined the channel height in the MD simulations as $H_0 - t$, where t is the effective thickness of one layer of solid atoms. When the channel height is comparable with the diameter of gas molecules, I believe that the effective height at the channel entry also depends on the size and incident angle of gas molecules. As the authors claim that they developed a generalized Knudsen theory, the authors should clarify that their theoretical model considers the entry effect in the case of the channel height comparable with size of gas molecules.

Reviewer #2 (Remarks to the Author):

The authors have answered all my questions in a convincing way. I recommend publication of the manuscript.

Reviewer #3 (Remarks to the Author):

The authors have properly addressed my concerns.

Point-by-Point Response to Reviewers' Comments

Reviewer #1 (Remarks to the Author):

I think the author addressed most of my concerns. But I still have one concern about the determination of channel height in their model. In the reply, the authors mentioned that they determined the channel height in the MD simulations as $H_0 - t$, where t is the effective thickness of one layer of solid atoms. When the channel height is comparable with the diameter of gas molecules, I believe that the effective height at the channel entry also depends on the size and incident angle of gas molecules. As the authors claim that they developed a generalized Knudsen theory, the authors should clarify that their theoretical model considers the entry effect in the case of the channel height comparable with size of gas molecules.

Response: Thanks. We agree with this viewpoint “when the channel height is comparable with the diameter of gas molecules, the effective height at the channel entry also depends on the size and incident angle of gas molecules”, and, accordingly, have now added some discussion in our revised manuscript.

Fig. R1. The gas flow rate in slits with varying heights at the angstrom scale, with a length of 10 nm and a width of 70 nm. The solid circles represent results from full-atom molecular dynamics (MD) simulations, the open circles correspond to results from virtual wall simulations, and the solid line shows the prediction from the generalized Knudsen theory. The error bars depict the standard deviation sourced from three distinct MD simulations.

This entry effect has been discussed in our previous work, see *Entry effects in Methods* of ref. [24]. The concept was illustrated in *Extended Data Fig. 7* of ref. [24]. We performed additional full-atom molecular dynamics (MD) simulations and virtual wall simulations, calculating gas flow rate through channels whose height is comparable with the diameter of gas molecules (less than 1.2 nm). We found that the entry effect plays a significant role only when the channel height is less than 0.3 nm, which is comparable to the size of argon gas (Fig. R1). On the other hand, the complication escalates when considering channels with different cross sections, such as circular, rectangular, or triangular. Therefore, integrating the incident angle into our current model presents significant challenges.

Reviewer #2 (Remarks to the Author):

The authors have answered all my questions in a convincing way. I recommend publication of the manuscript.

Response: Let us thank Reviewer for careful reading of our paper and useful comments.

Reviewer #3 (Remarks to the Author):

The authors have properly addressed my concerns.

Response: We are grateful to the Reviewer for the time spent on working with our manuscript and valuable comments, which did help us to improve the paper.